# Monkeypox in a Patient with Controlled HIV Infection Initially Presenting with Fever, Painful Pharyngitis, and Tonsillitis

**DOI:** 10.3390/medicina58101409

**Published:** 2022-10-07

**Authors:** Fabian Fischer, Alexander Mehrl, Melanie Kandulski, Sophie Schlosser, Martina Müller, Stephan Schmid

**Affiliations:** Department of Internal Medicine I, Gastroenterology, Hepatology, Endocrinology, Rheumatology and Infectious Diseases, University Hospital Regensburg, 93053 Regensburg, Germany

**Keywords:** monkeypox, fever, pharyngitis, tonsillitis, HIV

## Abstract

*Background and Objectives*: With more and more cases emerging outside central and west African countries, where the disease is endemic, the World Health Organization (WHO) has recently declared human monkeypox a Public Health Emergency of International Concern. Typical symptoms of the disease include fever, myalgia, and lymphadenopathy followed by a rash, but other symptoms may occur. Immunocompromised patients, including patients with uncontrolled Human Immunodeficiency Virus (HIV) infection, may be at risk for more severe courses. *Case presentation:* We present the case of a 30-year-old male patient of Brazilian descent with monkeypox. Initial symptoms were fever and general discomfort, with painful pharyngitis and tonsillitis and finally a papular rash of the anogenital area as the disease progressed. The presumed date of infection was a sexual contact with an unknown male eight days before the first symptoms occurred. The patient had a known and controlled HIV infection. The main reason for the initial presentation at the hospital was painful pharyngitis and tonsillitis, limiting food intake. Monkeypox infection was confirmed via PCR testing from a swab sample of cutaneous lesions. Adequate systemic and local analgesia enabled oral food uptake again. Antiviral therapy with Tecovirimat was not administered due to the stable immune status of the patient and the mild clinical symptoms. To cover a possible bacterial superinfection or Syphilis infection of the tonsil, antibiotic therapy with Ceftriaxone was added. Several days after presentation, the inflammation of the pharynx resolved and was followed by non-painful mucosal peeling. The patient was followed up with telephone calls and reported a complete recovery. The skin lesions were completely dried out 18 days after the first symptoms. *Conclusions:* Painful pharyngitis and tonsillitis can be rare early symptoms of monkeypox, which is highly relevant in everyday clinical practice. Particularly in patients with risk factors for monkeypox infection, further clinical and microbiologic testing for monkeypox should be performed if there is a clinical presentation with pharyngitis and tonsillitis.

## 1. Background

Human monkeypox is a zoonotic disease caused by the Monkeypox virus, an Orthopox virus of the Poxviridae family, thus related to the Variola virus, the causative agent of smallpox [1]. The reservoirs of the monkeypox virus include, in addition to monkeys, several rodents [2]. After initial transmission from animals to humans, spreading from human to human is possible through close physical contact, respiratory droplets, or fomite infection [3,4,5]. Previous publications report an incubation period of around 3–21 days [1,6]. Cases of monkeypox have been limited mainly to West and Central Africa. The first outbreak in the Western hemisphere appeared in 2003 in the United States of America and was ascribable to imported rodents [7]. Since May 2022, however, an increasing number of cases have emerged worldwide since May 2022. First analyses of the current outbreak report sexual contacts as the suspected main route of disease acquirement, with approximately 95% of total infections [1]. Although sexual activity has not been proven as a mode of transmission, the available data on the current outbreak support the concept [8]. For instance, a pooled analysis reported sexual exposure in 91.67% [9], and Monkeypox virus DNA could be detected in seminal fluid [1,10]. Men who have sex with men (MSM) constituted 98% of the patients [1]. Notably, around 40% of the infected patients lived with HIV [11].

The clinical presentation of monkeypox is characterized by a distinctive rash: It often begins with macula followed by papula. After the development of vesicular and finally pustular or pseudo-vesiculopustular phases, crusts appear, and lesions dry up [3,6,12]. The whole body can be affected. The anogenital area, face, or extremities are common anatomical sites [1]. Usually, there is a prodromal phase of fever, headache, malaise, and lymphadenopathy preceding the rash; lymphadenopathy especially can be characteristic of monkeypox in comparison to related diseases such as smallpox and chickenpox [3]. Of note, not every patient reported systemic prodromal symptoms in the current outbreak [1,3,4,6]. Most cases are mild and self-limiting, but relevant complications have been reported, including secondary bacterial infections, bronchopneumonia, sepsis, or encephalitis [3]. Immunocompromised patients, including patients with advanced HIV infection, may be prone to severe courses [4]. Of clinical relevance, a recently published case series noticed similar clinical features among people living with HIV under effective antiretroviral therapy (ART) and non-HIV patients [1]. To date, Tecovirimat is the only therapeutic option for monkeypox approved by the European Medicines Agency (EMA) [13]. This antiviral agent was originally developed for the treatment of smallpox, but showed efficacy against Monkeypox infection in animals [14] and has also been used in humans [15]. There are two available vaccines against smallpox, probably providing protection against monkeypox. The ACAM2000 vaccine consists of a live and replicating Vaccinia virus, whereas the newer JYNNEOS/Imvanex vaccine contains a non-replicating virus [16]. As of August 2022, there have been more than 30,000 confirmed monkeypox cases worldwide in a total of 88 countries and territories, according to the Center for Disease Control and Prevention (CDC). Of these, monkeypox has not been reported historically in 81 countries [17]. Previously, monkeypox had been declared an “evolving threat of moderate public health concern” by the World Health Organization on 23 July 2022 [18].

## 2. Case Presentation

Here, we report the case of a 30-year-old male patient referred to our tertiary care center on July 2022 with suspected monkeypox for further diagnostics and treatment. 

### 2.1. Recent Medical History

The patient originated from Brazil and was living in Germany for five years to attend university. He had been diagnosed with HIV seven years ago and was under ART with Emtricitabine/Tenofovir/Efavirenz since being diagnosed with HIV. Otherwise, no chronic medical conditions were known, and no long-term medication was taken. Eight days before symptom onset, he attended a pride parade and had sexual contact with an unknown male. He first noticed fever, myalgia and malaise, followed by painful pharyngitis and tonsillitis and finally a papular rash of the anogenital area on day four of the illness. Rectal pain or tenesmus was not reported. Due to a painful sore throat, oral food intake was no longer possible, leading to the initial presentation at the hospital. 

### 2.2. Presentation

Upon physical exam, the patient presented with stable vital parameters and was non-febrile. Lymphadenopathy could not be detected. Papulovesicular skin lesions could be found all over the body, with the highest density in the perineal and anal region (Figure 1a). Other papules affected the lower and upper extremities (Figure 1b) and the tip of the tongue (Figure 2). A homogenous, plane and markedly red enanthema were present in the pharyngeal region. Ulcers or vesiculae were absent in the pharynx. The left palatine tonsil appeared ulcerous and swollen.

### 2.3. Clinical Course

With suspected monkeypox, we established respective isolation measures. Polymerase chain reaction (PCR) testing from a swab sample of the anogenital lesions turned positive for Orthopox viridae, and further analysis confirmed monkeypox. Laboratory workup revealed elevated CRP without significant leukocytosis or procalcitonin. HIV viral load was undetectable, and CD4 counts were 600/µL. Shortly after initiating adequate systemic and local analgesia, oral food uptake was possible again. We decided against antiviral therapy with Tecovirimat due to the competent immune status of the patient and the mild presentation. We decided to cover possible bacterial superinfection or Syphilis infection of the tonsil by adding an antibiotic therapy with Ceftriaxone. 

### 2.4. Outcomes

After several days, inflammation of the pharynx decreased, and non-painful peeling of the mucosa developed (Figure 2). 

**Figure 2 medicina-58-01409-f002:**
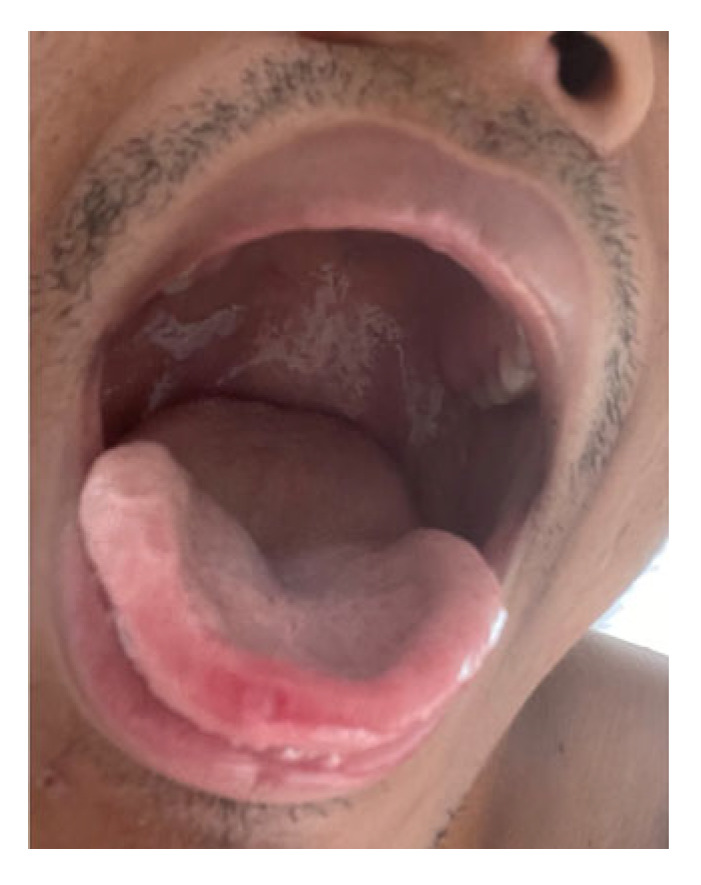
Peeling of the mucosa following acute pharyngitis and monkeypox-associated ulcer at the tip of the tounge.

We recommended isolation until complete recovery of the skin lesions upon discharge. Antibiotic therapy was discontinued when the tonsillitis improved and serology for *Treponema pallidum* was unsuggestive of acute infection. The patient reported a complete recovery when followed up by telephone calls. The skin lesions completely resolved 18 days after the first symptoms and quarantine was lifted by local authorities according to German regulations. The time course of the monkeypox infection is shown in Figure 3.

## 3. Discussion

We describe a case of monkeypox in an MSM patient with HIV and competent immune status under ART. Whereas information about the incubation period is yet indistinct, we were able to identify a sexual contact as the highly probable time point of infection, resulting in an incubation period of seven days. Upon presentation, the predominantly affected areas were the oral and anogenital regions. This clinical presentation is in accordance with—so far published—analyses of the current outbreak, identifying sexual transmission as the main route of infection and MSM as a major population at risk for monkeypox [1,11,19,20].

While people living with HIV seem to represent a significant proportion of current monkeypox patients, clinical courses appear similar among HIV positive and HIV negative cases, if HIV infection is well controlled [1,19,20]. Uncontrolled HIV infection, however, may pose a risk for severe and prolonged courses of monkeypox [4] and treatment with Tecovirimat should be considered in this case.

Of the two available vaccinations, the JYNNEOS/Imvanex vaccine was approved by EMA for prophylaxis of monkeypox on 22 July 2022 [21,22]. Two doses are required. German and international guidelines recommend vaccination for laboratory personnel working with Orthopox viridae and MSM ≥ 18 for pre-exposure prophylaxis [23]. Of note, more vaccine will be required to offer immunization to everyone belonging to this risk group. Therefore, it may be clinically reasonable to focus on the first dose and further risk-stratification. First vaccinations were administered in July 2022 in Germany. It remains to be eluded if HIV infection per se increases vulnerability for monkeypox or if there is a high overlap of populations at risk.

## 4. Conclusions 

Of clinical relevance, HIV caregivers should be highly aware of monkeypox, concerning signs and symptoms, as well as treatment and prophylaxis options, and readily provide information to their patients. Our case report shows that painful pharyngitis and tonsillitis can be rare early symptoms of monkeypox, which is highly relevant in everyday clinical practice.

## Figures and Tables

**Figure 1 medicina-58-01409-f001:**
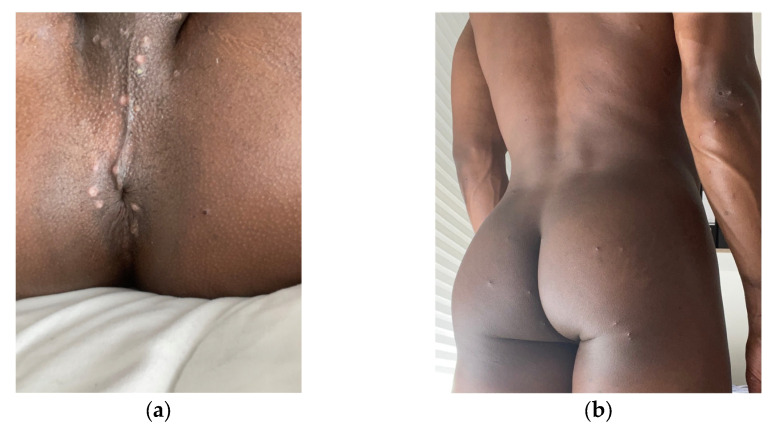
(**a**) Papular and vesicular skin lesions affecting the anal and perineal region. (**b**) Singular papules, vesiculae or pustules on glutes and arms.

**Figure 3 medicina-58-01409-f003:**
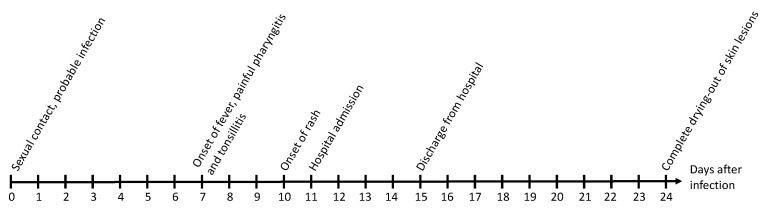
Time course of the monkeypox infection.

## Data Availability

The datasets generated and/or analyzed during the current study are not publicly available due to data privacy, but are available from the corresponding author on request.

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
