# Peer review of "Monkeypox in a Patient with Controlled HIV Infection Initially Presenting with Fever, Painful Pharyngitis, and Tonsillitis"

_medicina, 2022, doi:10.3390/medicina58101409_

Round 1

Reviewer 1 Report

1. Line 14, 15  It is still unknown if uncontrolled HIV is a risk factor for severe monkeypox infection since many individuals with HIV and monkeypox actually have well-controlled HIV / undetectable viral load.    2. In Figure 2 - a punched-out ulcer is visualized at the tip of the tongue. Was this a mucosal lesion of monkeypox? If yes please include in the figure legend    3. The authors mention mucosal involvement, especially pharyngitis, and tonsillitis at several points in the manuscript. It would be important to shed light on this enanthem due to monkeypox when the clinical presentation is discussed. An enanthem can be the sole and the only presenting feature of the disease.    4. The authors describe the involvement of the anal region in this case. Did the patient have any rectal pain proctitis, tenesmus, etc.? 

Author Response

Response to Reviewer 1:

Thank you very much for critical reading of our manuscript and valuable comments. We have addressed your criticisms carefully, have accepted all your suggestions and have made the appropriate changes in the text.

Reviewer 1 – Comment 1: Line 14, 15 It is still unknown if uncontrolled HIV is a risk factor for severe monkeypox infection since many individuals with HIV and monkeypox actually have wellcontrolled HIV / undetectable viral load.

Response: We absolutely agree with the reviewer. Uncontrolled HIV infection can only be suspected as a risk factor for severe monkeypox infections based on available evidence. The respective sentences in the abstract and discussion have been changed.

Reviewer 1 – Comment 2: In Figure 2 - a punched-out ulcer is visualized at the tip of the tongue. Was this a mucosal lesion of monkeypox? If yes, please include in the figure legend.

Response: Thank you for the valuable advice. The punched-out ulcer in Figure 2 was indeed a suspected monkeypox lesion, as it is now stated in the figure legends.

Reviewer 1 – Comment 3: The authors mention mucosal involvement, especially pharyngitis, and tonsillitis at several points in the manuscript. It would be important to shed light on this enanthem due to monkeypox when the clinical presentation is discussed. An enanthem can be the sole and the only presenting feature of the disease.

Response: The reviewer is addressing a very important issue here. We included the clinical description of the pharyngeal enanthema in “Case presentation”.

Reviewer 1 – Comment 4: The authors describe the involvement of the anal region in this case. Did the patient have any rectal pain proctitis, tenesmus, etc.?

Response: The patient did not report any rectal pain, which is now included in “Recent medical history”.

Changes in the text and in the figures:

Introduction and discussion revised; Figure legend 2 revised, case presentation and recent medical history stated more precisely.

Reviewer 2 Report

In my opinion the paper was clear, readable and informative and will provide a valuable source document for anyone requiring a primer to know and understand this issue. Some changes are needed:        
  • Lines 16-29: Consider avoiding writing in the first person way. The paragraph `Case presentation` should be reconstructed in a way that there is no repetition with the text of the manuscript. Have the repetition be minimal.
  • Line 37: The cause of human monkeypox is a member of the Orthopoxvirus genus of the Poxviridae family. Specify the exact name of the virus, it is not enough to state Ortopoxvirus.  
  • Lines 42-44: State an appropriate reference. Rationale: the previously cited reference No. 6 was published in 2019, therefore it cannot refer to the following sentences.
  • Line 44: This is not precise enough: the first outbreak of human monkeypox in western hemisphere occurred in 2003. 
  • Lines 44-46: Cite an appropriate reference for the sentence on these lines.   
  • Line 45: Cite the reference which confirms whether sexual transmission as mode of transmission human monkeypox is confirmed or not. 
  • Line 49: In order to point to the meaning of this case report, you must describe the entire course of the disease (you must describe the prodromal phase, etc.). 
  • Lines 49-51: This must be written more precisely and appropriate references must be cited. For example, the rash involves not only `papular, vesicular, or pustular rash`, but also includes macules, papular, vesicular and pustular phases, and finally crusts. All three sentences must be written more clearly and more detailed.  
  • Line 58: Explain the abbreviation EMA. 
  • Line 66: Is the CDC abbreviation necessary, since `the Center for Disease Control and Prevention` are mentioned only in this place in the text.   
  • Line 68: The World Health Organization is mentioned only once in the text of the paper, therefore the WHO abbreviation is not necessary. 
  • Line 68: State the exact date for when the declaration was made. Also, the stated year is incorrect.
  • Line 71: Provide the date of when the patient came to your healthcare institution.
  • Lines 82-87: State whether lymphadenopathy was found in this patient or not. If yes - state the localization.
  • Line 107: State the full name, instead of `T. pallidum`. 
  • Line 116: Be precise - is the sexual contact the "cause" or the circumstance that facilitated the transmission of infection. 
  • Lines 131-132: State when the use of vaccine began in Germany during the current outbreak. Also, specify the indications for this vaccination in Germany. 
  • Table on page 4: Explain all abbreviations in the text when they are first used, in line with the instructions for authors. 
  • Table on page 4: CDC - not necessary, delete (see above mentioned).
  • Table on page 4: Does the text of this paper even mention COVID-19 as an abbreviation? No, so delete from this list of abbreviations.
  • Table on page 4: PCR should be corrected.  
  • Table on page 4: WHO - not necessary, delete (see above mentioned).
  • Line 173: Cite this reference No. 7 according to the Instructions for Authors.  
  • Line 174: Cite this reference No. 8 according to the Instructions for Authors.  
  • Line 175: Cite this reference No. 9 according to the Instructions for Authors.   
  • Line 181: Cite this reference No. 12 according to the Instructions for Authors.    
  • Line 182: Cite this reference No. 13 according to the Instructions for Authors.     
  • Line 183: Cite this reference No. 14 according to the Instructions for Authors.    
  • Line 190: Cite this reference No. 17 according to the Instructions for Authors.   
  • Line 191: Cite this reference No. 18 according to the Instructions for Authors.    
  • Line 192: Cite this reference No. 19 according to the Instructions for Authors.      

Author Response

Thank you very much for your very valuable comments. We have addressed all your criticisms carefully, have accepted all your suggestions and have made the appropriate changes in the text.

Reviewer 2 – Comment 1: Lines 16-29: Consider avoiding writing in the first-person way. The paragraph “Case presentation” should be reconstructed in a way that there is no repetition with the text of the manuscript. Have the repetition be minimal.

 Response: We thank the Reviewer for pointing out this issue. We have revised the abstract and the paragraph “case presentation” to avoid repetitions.

Reviewer 2 – Comment 2: Lines 44-46: Cite an appropriate reference for the sentence on these lines AND Line 45: Cite the reference which confirms whether sexual transmission as mode of transmission human monkeypox is confirmed or not AND Line 116: Be precise - is the sexual contact the "cause" or the circumstance that facilitated the transmission of infection.

Response: We thank the reviewer for this important comment. Sexual transmission of monkeypox is not yet proven although available data highly suggest this transmission route. The respective parts have been adapted and citation has been added.

Reviewer 2 – Comment 3: Line 49: In order to point to the meaning of this case report, you must describe the entire course of the disease (you must describe the prodromal phase, etc.) AND Lines 49-51: This must be written more precisely, and appropriate references must be cited. For example, the rash involves not only `papular, vesicular, or pustular rash`, but also includes macules, papular, vesicular and pustular phases, and finally crusts. All three sentences must be written more clearly and more detailed AND Lines 82-87: State whether lymphadenopathy was found in this patient or not. If yes - state the localization.

Response: Thank you for noting. We added more detail and references to the description of clinical features of monkeypox in general and in our patient.

Reviewer 2 – Comment 4: Line 58: Explain the abbreviation EMA AND Line 66: Is the CDC abbreviation necessary, since `the Center for Disease Control and Prevention` are mentioned only in this place in the text AND Line 68: The World Health Organization is mentioned only once in the text of the paper, therefore the WHO abbreviation is not necessary AND Line 107: State the full name, instead of `T. pallidum`.

Response: Thank you. These abbreviations have been corrected or deleted if unnecessary.

Reviewer 2 – Comment 5: Line 68: State the exact date for when the declaration was made. Also, the stated year is incorrect. AND Line 71: Provide the date of when the patient came to your healthcare institution. AND Lines 131-132: State when the use of vaccine began in Germany during the current outbreak. Also, specify the indications for this vaccination in Germany.

Response: We absolutely agree. The respective dates have been provided.

Reviewer 2 – Comment 6: Line 173: Cite this reference No. 7 according to the Instructions for Authors etc.

Response: Thank you for noticing. The references have been aligned to information for Authors. We apologize for missing this in the first place.

Changes in the text:

Introduction revised and further reference added, abstract and case presentation revised, references added where correctly noted, abbreviations adapted, references aligned.

Round 2

Reviewer 1 Report

Thank you for the significantly improved revision. 

Authors may consider the addition/ inclusion of "pseudo-vesiculopustules" to listed cutaneous morphologies seen in monkeypox. 

Khanna U, Kost Y, Wu B. Diagnostic Considerations in Suspected Cases of Monkeypox. J Am Acad Dermatol. 2022 Sep 24:S0190-9622(22)02777-3. 

Author Response

We thank the reviewer for the important advice to include the article "Diagnostic Considerations in Suspected Cases of Monkeypox" by Khans et al. We have included this very relevant article in our revised manuscript. Once again  thank you for your very precious advice, which is highly appreciated. 

Reviewer 2 Report

The authors have done a good job in addressing all of the points that I raised regarding their submission. Thanks.  

Author Response

Thank you for your very precious advice regarding our manuscript, which is highly appreciated by me and all the coauthors. Your comments substantially improved the manuscript.